# Glyoxylate Shunt and Pyruvate-to-Acetoin Shift Are Specific Stress Responses Induced by Colistin and Ceragenin CSA-13 in *Enterobacter hormaechei* ST89

Suhanya V. Prasad,[a] Krzysztof Fiedoruk,[a] Magdalena Zakrzewska,[a] Paul B. Savage,[b] Robert Bucki[a]

[a]Department of Medical Microbiology and Nanobiomedical Engineering, Medical University of Bialystok, Bialystok, Poland
[b]Department of Chemistry and Biochemistry, Brigham Young University, Provo, Utah, USA

**ABSTRACT** Ceragenins, including CSA-13, are cationic antimicrobials that target the bacterial cell envelope differently than colistin. However, the molecular basis of their action is not fully understood. Here, we examined the genomic and transcriptome responses by *Enterobacter hormaechei* after prolonged exposure to either CSA-13 or colistin. Resistance of the *E. hormaechei* 4236 strain (sequence type 89 [ST89]) to colistin and CSA-13 was induced *in vitro* during serial passages with sublethal doses of tested agents. The genomic and metabolic profiles of the tested isolates were characterized using a combination of whole-genome sequencing (WGS) and transcriptome sequencing (RNA-seq), followed by metabolic mapping of differentially expressed genes using Pathway Tools software. The exposure of *E. hormaechei* to colistin resulted in the deletion of the *mgrB* gene, whereas CSA-13 disrupted the genes encoding an outer membrane protein C and transcriptional regulator SmvR. Both compounds upregulated several colistin-resistant genes, such as the *arnABCDEF* operon and *pagE*, including genes coding for DedA proteins. The latter proteins, along with beta-barrel protein YfaZ and VirK/YbjX family proteins, were the top overexpressed cell envelope proteins. Furthermore, the L-arginine biosynthesis pathway and putrescine-ornithine antiporter PotE were downregulated in both transcriptomes. In contrast, the expression of two pyruvate transporters (YhjX and YjiY) and genes involved in pyruvate metabolism, as well as genes involved in generating proton motive force (PMF), was antimicrobial specific. Despite the similarity of the cell envelope transcriptomes, distinctly remodeled carbon metabolism (i.e., toward fermentation of pyruvate to acetoin [colistin] and to the glyoxylate pathway [CSA-13]) distinguished both antimicrobials, which possibly reflects the intensity of the stress exerted by both agents.

**IMPORTANCE** Colistin and ceragenins, like CSA-13, are cationic antimicrobials that disrupt the bacterial cell envelope through different mechanisms. Here, we examined the genomic and transcriptome changes in *Enterobacter hormaechei* ST89, an emerging hospital pathogen, after prolonged exposure to these agents to identify potential resistance mechanisms. Interestingly, we observed downregulation of genes associated with acid stress response as well as distinct dysregulation of genes involved in carbon metabolism, resulting in a switch from pyruvate fermentation to acetoin (colistin) and the glyoxylate pathway (CSA-13). Therefore, we hypothesize that repression of the acid stress response, which alkalinizes cytoplasmic pH and, in turn, suppresses resistance to cationic antimicrobials, could be interpreted as an adaptation that prevents alkalinization of cytoplasmic pH in emergencies induced by colistin and CSA-13. Consequently, this alteration critical for cell physiology must be compensated via remodeling carbon and/or amino acid metabolism to limit acidic by-product production.

**KEYWORDS** colistin, ceragenins, CSA-13, transcriptional regulation, adaptive resistance, next-generation sequencing, stress response, carbon metabolism, CAMPs, RNA-seq

Address correspondence to Krzysztof Fiedoruk, krzysztof.fiedoruk@umb.edu.pl, or Robert Bucki, buckirobert@gmail.com.

The authors declare a conflict of interest. Robert Bucki received funding from N8 Medical, which focuses on commercialization of medical devices that incorporate ceragenins, to attend TechnoSepsis meeting on Oct 5-7, 2022 in Spain.

Colistin was one of the first antibiotics discovered and is nearly as old as antimicrobial therapy. However, its recent reemergence as the "last resort" drug against carbapenem-resistant bacteria revealed our misunderstanding of its properties, even at the level of antibiotic susceptibility testing (AST) (1–3).

Colistin is problematic in testing due to the variety and constantly evolving (hetero) resistance mechanisms and paradoxical effects, such as "skipped well" or species-dependent increasing/decreasing MICs in the presence of divalent cations (2, 4). Therefore, recent findings connecting resistance to colistin with globally regulated physiological processes, such as ion homeostasis, membrane potential, respiration, and carbon metabolism, regardless of lipopolysaccharide (LPS) modification (e.g., by L-Ara4N), are very engaging (5–7). Notably, a study by Panta et al. demonstrated a universal relationship between colistin activity and cytoplasmic pH homeostasis, likely mediated by proton and non-proton pump systems (5). Although the molecular basis of this phenomenon remains to be elucidated, metabolic processes preventing intracellular alkalization, such as fermentation, are certainly implicated (5, 7). Remarkably, in this scenario, the proton motive force (PMF) may directly impact interactions between colistin and LPS due to remodeled metabolism (8).

In light of this, the purpose of the present research was to determine whether another cationic agent, CSA-13, a member of the ceregenin family of antimicrobials, may induce similar behaviors in bacteria. The properties of ceragenins as broad-spectrum and microbicidal agents, along with their distinctive mode of action unrelated to cross-resistance to other antimicrobials, are well acknowledged (9–12). Therefore, it might be assumed that the scope of their activity comprises structures or processes universal across bacterial species. The ESKAPE group (*Enterococcus faecium*, *Staphylococcus aureus*, *Klebsiella pneumoniae*, *Acinetobacter baumannii*, *Pseudomonas aeruginosa*, and *Enterobacter* species) pathogen *Enterobacter hormaechei* of sequence type 89 (ST89) was used as a target organism. The selection is not coincidental since, in Poland, it is a growing problem, in particular in the context of carbapenem resistance (13).

## RESULTS

**Induced resistance to colistin and CSA-13.** Induction of resistance in *E. hormaechei* strain 4236 to colistin (denoted here as the Eh4236ColR isolate) and CSA-13 (denoted here as the Eh4236Csa13R isolate) by serial-passage experiments resulted in a high level of resistance to colistin: i.e., an increase in the MIC for colistin in Eh4236ColR from 0.125 mg/L to 128 mg/L. In contrast, the final MIC for CSA-13 in Eh4236Csa13R varied between 32 and 64 mg/L compared to the initial MIC of 4 mg/L.

**Colistin- and CSA-13-mediated changes at genomic level.** Whole-genome sequencing (WGS) of *E. hormaechei* 4236 isolates (i.e., the wild type and those with induced resistance to colistin and CSA-13) allowed for their comparative analysis and identification of genome-level changes. In detail, the genome of the studied *E. hormaechei* 4236 strain is composed of a chromosome (4,689 coding DNA sequences [CDSs]) and one plasmid (129 CDSs). However, virtually all changes were observed at the level of the chromosome. In detail, deletion of the 5,028-bp chromosomal region involving the *mgrB* gene, encoding a negative regulator of the PhoPQ signaling system, which is directly responsible for developing resistance to colistin in *Enterobacter cloacae* complex and other Gram-negative rods (14, 15), was present in the Eh4236ColR isolate. In contrast, the genes encoding outer membrane (OM) protein C (OmpC) and transcriptional regulator SmvR were disrupted by the insertion of premature stop codons in Eh4236Csa13R. Furthermore, an ~28-kb fragment, a putative mobile element equipped with IS*5* transposase and Arm-type integrase utilizing tRNA-Leu as the integration site, was deleted from the chromosome of this isolate. None of the genes in this region were previously known to be associated with colistin resistance or to be abnormally expressed in Eh4236ColR isolate; therefore, this region appears to be irrelevant for the development of colistin resistance. The deleted and/or disrupted genes were excluded from the further transcriptomic analysis (see below).

**Colistin and CSA-13 mediated changes at the transcriptomic level.** Transcriptome sequencing (RNA-seq) analysis of *E. hormaechei* 4236 isolates (i.e., the wild type and those with induced resistance to colistin and CSA-13) allowed for their comparative analysis and identification at the transcriptomic level (i.e., differentially expressed genes [DEGs]). In the

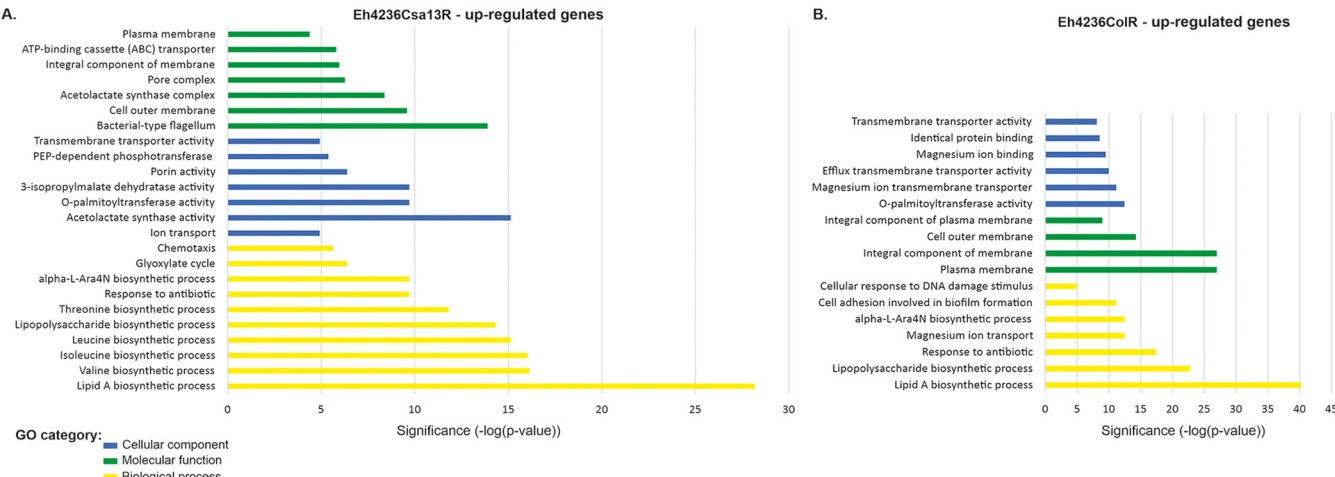

**FIG 1** Gene Ontology (GO) categories of the upregulated genes in Eh4236Csa13R (A) and Eh4236ColR (B).

CSA-13-resistant strain, more extensive transcriptional changes were observed, in the number and variety of DEGs as well as their vitality for the cell. Overall, 6.1% and 2.4% of the total gene contents, representing multiple categories from ribosomal proteins to flagellar components (Fig. 1 and Fig. 2), were affected in Eh4236Csa13R and Eh4236ColR, respectively. In addition, the ratio between up- and downregulated genes—133 (mean, 3.9; range, 2.0- to 26.4-fold change) versus 151 (mean, 3.9; range, 2.0- to 32.4-fold change)— was similar in Eh4236Csa13R, whereas the overexpressed genes clearly predominated—71 (mean, 10.4; range, 2.0- to 55.8-fold change) versus 40 (mean, 4.6; range, 2.2- to 19.2-fold change) ($P = 0.0024$, two-tailed Fisher exact test) genes underexpressed—in the colistin-treated isolate. Furthermore, both isolates shared 33 upregulated genes. However, their expression was significantly higher ($P = 0.0023$; Student's *t* test) in Eh23465ColR. On the contrary, the downregulation level of the 17 shared genes was, in general, more visible in Eh4236CsaR, with the notable exception of the putrescine-ornithine antiporter (*potE-speF*) operon, but statistically irrelevant (see Tables S1 to S4 in the supplemental material). Finally, metabolic mapping linked the dysregulated genes with 54 pathways (17 superpathways) and 32 pathways (7 superpathways), in Eh4236Csa13R and Eh4236ColR, respectively (see Fig. S1 and S2 in the supplemental material). Functional characteristics and statistics of DEGs are indicated in Fig. S3 to S15 in the supplemental material.

**Variation of the cell envelope transcriptome associated with cationic agent resistance.** Both tested agents elevated the expression of multiple genes associated with cell envelope synthesis and turnover (e.g., PBP6 [DacC], LPS assembly protein A [lapA], or

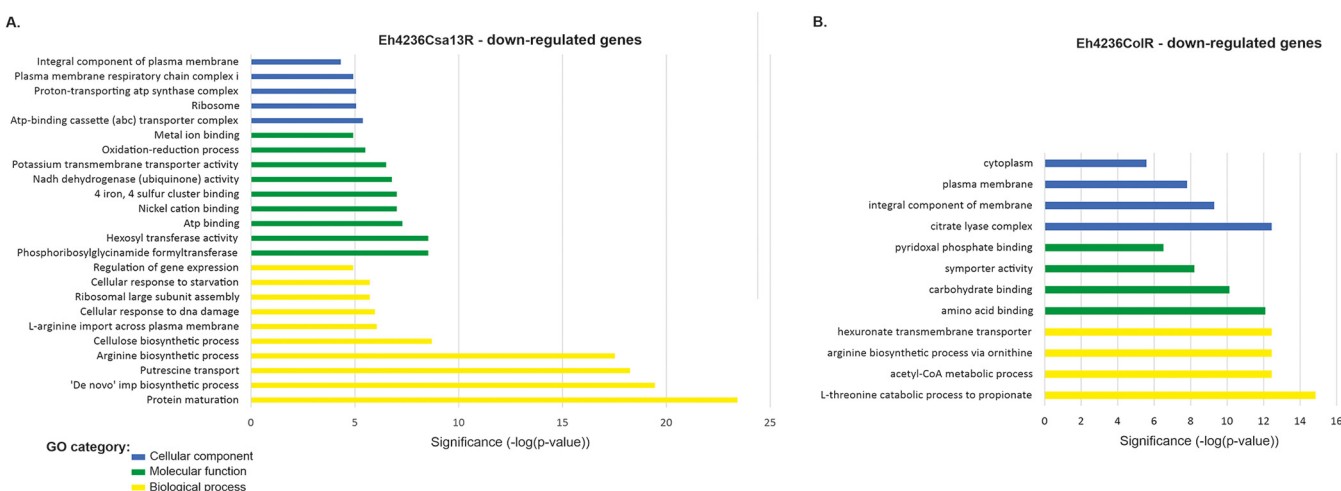

**FIG 2** Gene Ontology (GO) categories of the downregulated genes in Eh4236Csa13R (A) and Eh4236ColR (B).

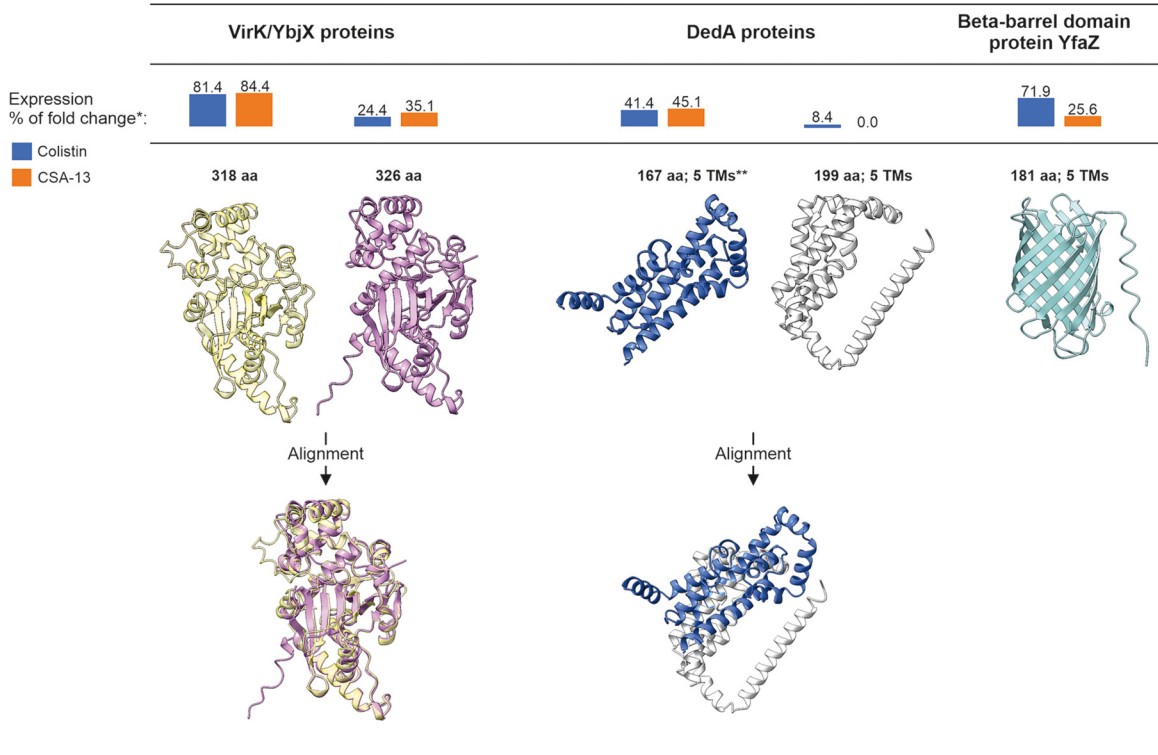

**FIG 3** Comparison of the three top overexpressed membrane proteins.

phosphatidic acid phosphatase [YbjG]), including members of the categories (i) cationic antimicrobial peptide (CAMP) resistance (ko01503) and (ii) response to antibiotics (GO:0046677). In the latter group, and in general, the *arnABCDEF* operon and *pagP* were the top overexpressed genes, with on average 37- and 14-fold changes of expression in the colistin- and CSA-13-resistant isolates, respectively. In Eh4236ColR, this mechanism is possibly augmented by the upregulated *pagABCD* operon (2.6-fold change). In contrast, the downregulation of phosphoethanolamine transferase (EptB) was an Eh4236Csa13R-specific trait. Furthermore, both isolates overexpressed multiple members in the categories response to antibiotics (GO:0046677) and cationic antimicrobial peptide (CAMP) resistance (ko01503), including outer membrane protein (OmpX), lipoprotein SlyB, undecaprenyl-diphosphatase (YbjG), and magnesium transporters (MgtA and MgtE), cation-transporting ATPase (CtpF), and multidrug export protein (MdtEF). Likewise, DedA family transporter (YabI) (16) was upregulated in Eh4236ColR and Eh4236Csa13R (Fig. 3). Nevertheless, a periplasmic protein, VirK/YbjX (318 amino acids [aa]) was the most abundantly expressed component of the membrane proteome, approaching the *arnABCDEF* operon's expression level. Moreover, another VirK/YbjX variant (326 aa) was also upregulated in both isolates (Fig. 3). Finally, the exposure to colistin and CSA-13 induced expression of the beta-barrel domain outer membrane (OM) protein YfaZ, especially in Eh4236ColR (Fig. 3). On the background of this dysregulated proteome, two Big-1 (bacterial immunoglobulin-like domain 1) family proteins are distinguished by their enormous size: 6,001 aa (29 Ig-like domains) and 3,807 aa (24 Ig-like domains). Interestingly, although the larger variant of Big-1 was upregulated in both isolates, particularly in Eh4236Csa13R (5.2-fold change), the smaller Big-1 variant was simultaneously suppressed to a corresponding degree in this isolate.

**Variation of the remaining transcriptome.** Significant downregulation of the *yjiY* gene (32.4-fold change) in Eh4236Csa13R and overexpression of the *yhjX* gene (32.6-fold change) in Eh4236ColR clearly differentiated both isolates. Pyruvate content is a potential signal of carbon and nitrogen availability inside the cell, and both two-component systems (TCSs) contribute to the regulation of carbon metabolism and amino acid biosynthesis via

L-glutamate (17, 18). Therefore, dysregulation of these transporters is possibly linked with a relatively large number of abnormally expressed genes involved in these processes (Fig. 1 and 2). To demonstrate their possible impact on certain metabolic processes and for clarity, we address this section in the Discussion.

## DISCUSSION

In a series of recent studies, Panta et al. demonstrated that maintaining an acidic cytoplasmic pH is essential for colistin resistance and that an alkaline pH acts synergistically with this polymyxin to suppress resistance in a variety of bacteria, including extremely resistant species, such as *Burkholderia thailandensis* and *Serratia marcescens* (5, 6, 19, 20). This process, however, may be compensated for by lowering the oxygen concentration (hypoxia) or by adding fermentable carbohydrates such as glucose, likely through acidifying products of the metabolic pathways activated under these conditions (5). In addition, the process is presumably mediated by proton and non-proton pump systems, so when the latter pumps are inhibited by colistin, the cell can only rely on proton pump systems, resulting in a progressive alkalinization of the cytoplasmic pH if is not compensated by an increase in proton uptake or a switch to hypoxic metabolism or fermentation (5, 21).

Indeed, our transcriptome study of *E. hormaechei* 4236 strain (ST89) is consistent with these findings, as both agents dysregulated expression of a number of genes involved in (i) carbon and (ii) amino acid metabolism, along with (iii) their transport systems, as well as (iv) proton and non-proton pump systems. Moreover, downregulation in both isolate genes involved in acid stress response (i.e., the arginine biosynthesis pathway and putrescine-ornithine antiporter PotE) suggests pH-related action of both agents, that triggers similar counteracting responses. According to the "alkaline stress" theory, we hypothesize that repression of the acid stress response, which alkalinizes cytoplasmic pH through amino acid decarboxylation and secretion of $H^+$, may be interpreted as an adaptation that diminishes these processes under emergency situations invoked by colistin and CSA-13. Consequently, this crucial cell physiology alteration must be balanced by modifying carbon and/or amino acid metabolism to limit the production of acidic by-products. Therefore, different patterns of abnormally expressed genes involved in carbon metabolism in both isolates possibly reflect more intense stress exerted by CSA-13, which was compensated for by repression of additional systems related to acid response, such as putrescine-specific $H^+$/symporter PotFGHI and formate hydrogen lyase (FHL) complex. Ultimately, the expression patterns of carbon and amino acid metabolism genes suggest that the stress exerted by colistin and CSA-13 was compensated for by remodeling carbon metabolism toward (i) fermentation of pyruvate to (*R*)-acetoin and (ii) glyoxylate pathway (glyoxylate shunt [GS]) supported by upregulated branched-amino-acid biosynthesis pathways in colistin- and CSA-13-resistant isolates, respectively (Fig. 4).

The putrescine-ornithine antiporter PotE, encoded in one operon, with ornithine decarboxylase SpeF, seems to be the central point in this acid response suppression network. An acidic environment activates SpeF, which converts L-ornithine to putrescine, consuming protons and generating membrane potential required for putrescine⇌ornithine exchange, as well as provides $CO_2$ for nucleotide synthesis (18, 22). Compared to other *Enterobacterales*, putrescine in *Enterobacter* spp. is produced in abundance (23), and the absence in this genus of glutamate decarboxylase (GadAB), a key decarboxylase in the *Escherichia coli* acid resistance response (24), implicates SpeF as its equivalent. Also, downregulation of the putrescine-specific $H^+$/symporter PotFGHI in Eh4236Csa13R and upregulation of the spermidine-acetyltransferase (SpeG), acetylating putrescine to *N*-acetylputrescine, is possibly a "putrescine-limiting" mechanism.

Depletion of putrescine as a source of L-glutamate and L-ornithine, along with repression of L-arginine, L-ornithine, and L-glutamate transporters (e.g., ArgM, ArtJ, and ArtM), is possibly coupled to downregulated genes involved in L-arginine biosynthesis: i.e., from L-glutamate via L-ornithine and L-citruline (L-arginine biosynthesis I). Moreover, in Eh4236Csa13R, two enzymes, ornithine carbamoyltransferase ArgF (EC 2.1.3.3) and carbamoyl phosphate synthase (CPSase) CarA (EC 6.3.4.14), responsible for production of carbamoyl phosphate (CP),

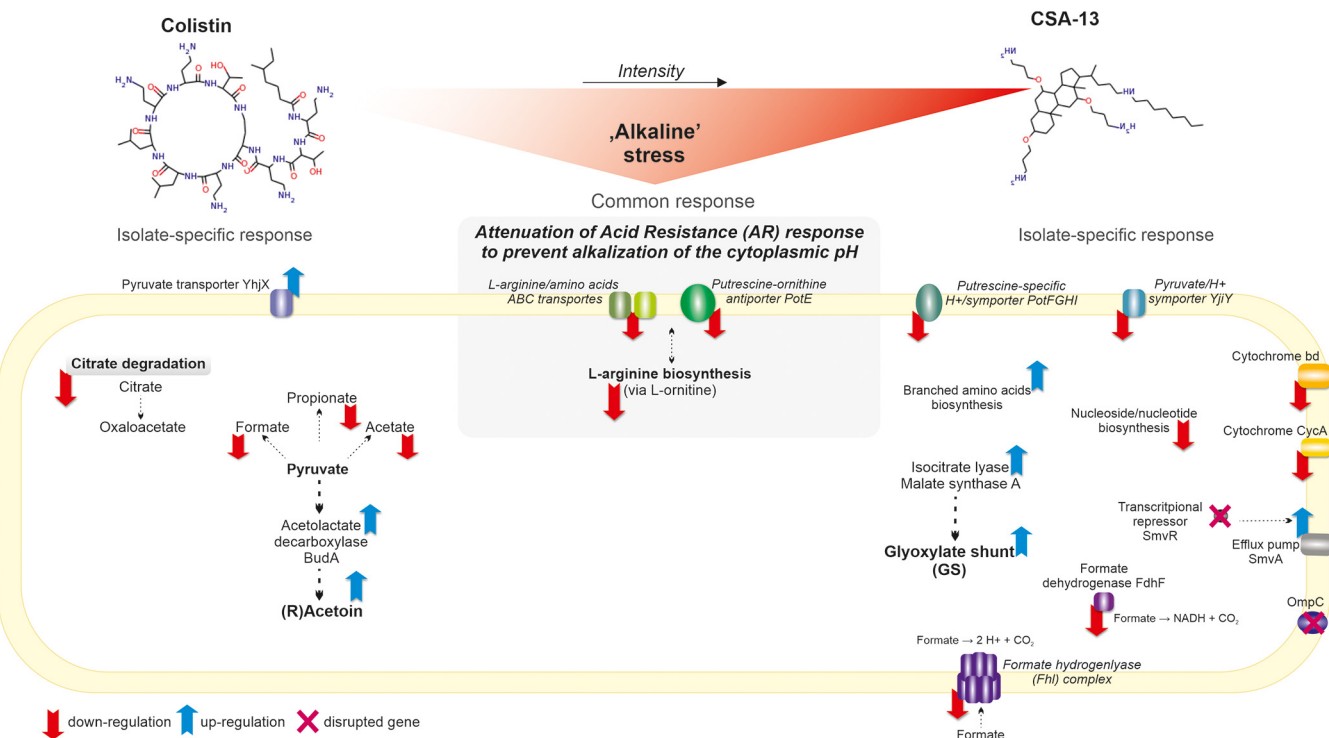

**FIG 4** Overview of the metabolic responses of the *E. hormaechei* 4236 strain to prolonged exposure to colistin and CSA-13.

were repressed. Since CPSase also provides CP for nucleotide/nucleoside biosynthesis (25), this may explain the downregulation of these genes in this isolate and may be a specific feature of ceragenin-induced stress (11). In contrast, the production of pyrimidine bases and D-ribose 5-phosphate appears to be enhanced by upregulated nucleotide 5′-monophosphate nucleotidase PnpN (EC 3.2.2.10) in the Eh4236ColR isolate.

Since pyruvate serves as an indicator of carbon and nitrogen availability in the cell (17, 18), two dysregulated pyruvate transporters, YhjX and YjiY, may link the attenuated acid response with carbon metabolism. Both transporters are controlled by similar and functionally interconnected pyruvate-sensing two-component systems (TCSs)—PyrSR (regulator of pyruvate reutilization formerly [formerly YpdAB]) belonging to the major facilitator superfamily (MFS) (17), and BtsSR (formerly YehUT), a CstA-like transporter (18, 26, 27)—but have different affinities to pyruvate. The BtsSR is a high-affinity receptor (sensing pyruvate at a concentration of >50 $\mu$M), whereas pyruvate concentrations of >600 $\mu$M activate the low-affinity protein PyrSR. Hence, the BtsSR system appears to be important under nutrient-limited conditions, specifically serine; thus, YjiY is a so-called "carbon starvation" transporter (18). For instance, the constitutive expression of *yjiY* that evolved in *E. coli* C41, a strain optimized for protein overproduction, possibly represents an adaptation toward efficient nutrient recovery, maintaining high productivity regardless of the nutrients' concentration (28). Overall, a cross talk between these TCSs was proposed as a mechanism ensuring fine-tuned pyruvate reutilization (17). Furthermore, PyrSR-mediated repression of other genes, including those involved in glutamate biosynthesis and arginine degradation, as well as those encoding membrane and stress proteins, has been recently reported by Miyake et al. (17).

Accordingly, the patterns of abnormally expressed genes involved in pyruvate metabolism appear to reflect distinct stress response strategies induced by colistin and CSA-13 (Fig. 4). Specifically, in Eh4236ColR, a tendency to reduce the formation of acidic products (i.e., formate, acetate, and propionate) was observed, which may explain the attenuation of the acetate-producing citrate operon. This "deprivation" of compounds driving essential metabolic pathways is possibly compensated by overexpression of acetolactate decarboxylase BudA (EC 4.1.1.5), which catalyzes pyruvate fermentation to (*R*)-acetoin, followed by acetyl coenzyme A (acetyl-CoA) and acetyl aldehyde production, and can be directly

incorporated into aerobic and fermentative pathways. Indeed, this strategy of pH homeostasis has been described in *Lactobacillus* (29), and cloning the acetoin pathway into bacteria lacking this pathway, such as *E. coli*, reduces the acidification process through formate hydrogen lyase (FHL) complex (30). Remarkably, in Eh4236Csa13R, another acetoin synthetase, (*S*)-acetoin, forming diacetyl reductase (EC 1.1.1.304), and the FHL complex were substantially downregulated. Furthermore, since acetoin at high concentrations can cause DNA and protein damage (31), upregulation by Eh4236ColR of universal stress protein A2 (UspA2) might be a counteractive response.

In contrast, with the CSA13-resistant isolate, formate synthesis appears to be slightly enhanced under anaerobic conditions by upregulation of formate C-acetyltransferase glycine radical (EC 2.3.1.54), and the most significant change is associated with the repression of two enzymes: formate dehydrogenase FdhF (EC 1.17.1.9), which oxidizes formate to $CO_2$ and NADH, and formate hydrogen lyase (FHL). The latter is a component of the FHL complex, a large, membrane-anchored system involving 16 proteins that takes up formate under anaerobic/fermentative conditions and acidic pH (32). Notably, none of these 16 genes, downregulated in Eh4236Csa13R, was affected in the colistin-resistant isolate. Moreover, the FHL complex produces the majority of $H^+$ from glucose fermentation (formate→$2 H^+ + CO_2$). The proton gradient generated in this process links the FHL complex with two non-proton pump systems downregulated in Eh4236Csa13R: i.e., the cytochrome *bd*-I oxidase subunit CydX and cytochrome CycA (33). Interestingly, this response interacts with the increased expression of cytochrome *bd* under alkaline stress in *E. coli*, along with the repression of proton pump respiratory chain complexes to minimize cytoplasmic proton loss during PMF generation (34). Consequently, the glyoxylate shunt (GS), which reduces the flow of electrons channeled into respiration during the tricarboxylic acid (TCA) cycle, driven by upregulated isocitrate lyase (EC 4.1.3.1) and malate synthase A (EC 2.3.3.9), is a potential carbon metabolic strategy adopted by this isolate. The GS avoids the carbon dioxide-generating stages of TCA and is the primary pathway under stress conditions, including oxidative, antibiotic, and cold/heat stresses (35, 36). In addition, the upregulation of the branched-amino-acid biosynthesis pathway that generates by-products, such as L-glutamate and 2-oxoglutarate, may be an essential part of metabolic strategy adopted by this isolate. However, an alternative, pH-related, explanation may be associated with the fact that inhibition of the first enzyme in this pathway, acetolactate synthase (EC 2.2.1.6), results in intracellular acidification (37).

Clearly, the specific responses of the isolates were associated with a distinct up- or downregulation of a number of cell transporters, notably efflux pumps. In general, these changes are more frequent in the colistin-resistant isolate. Notably, upregulation of the efflux pump SmvA in Eh4236Csa13 is presumably due to the disruption of its transcriptional repressor SmvR. SmvA and SmvR have been associated with decreased sensitivity or resistance to other cationic compounds, such as chlorhexidine and octenidine, in various bacteria (38–40). In addition, there was a substantial difference between upregulation (in Eh4236ColR) and downregulation (in Eh4236Csa13) of $K^+$ transporter (KdpABCDE) subunits. In contrast, the downregulation of the autoinducer-2 ABC transporter and the remaining genes of this operon (*lsrABCRK*) was a common but unanticipated characteristic.

Alteration of LPS by L-Ara4N residues, as a mechanism mediated by proton motive force (PMF)-dependent transporters, should be perceived in light of the aforementioned observations. In fact, L-Ara4N incorporation into LPS seems to be another physiological response to acidic stress coordinated by DedA family transporters (5, 6, 19, 41). Indeed, the DedA transporter was overexpressed in both isolates at comparable levels (Fig. 3). In addition, another *dedA* gene, encoding a DedA protein variant with 22 additional residues, was upregulated in Eh4236ColR. Nonetheless, the most overexpressed protein was periplasmic VirK/YbjX (318 aa), a member of the family of proteins of unknown function (DUF535). Moreover, another VirK/YbjX variant (326 aa) was also overexpressed in both isolates (Fig. 3). Initially, these proteins were identified with *Shigella flexneri* virulence factor VirK (42). However, VirK and YbjX should rather be perceived as membrane stress chaperons, preventing misfolding and aggregation of unstructured proteins (43, 44). Interestingly, in *Salmonella enterica* VirK has been recognized as essential for resistance to polymyxin B and other CAMPs (45).

The role of the mutation in Eh4236Csa13R OmpC, one of the key porins, in CSA-13 resistance may be analyzed from many perspectives, such as its role in maintaining membrane asymmetry or in binding $Mg^{2+}$ (46–48). However, the fact that the presence of OmpC reduces the ability to survive at alkaline pH may be valid in this context (49).

In conclusion, LPS modifications mediated by ArnABCDEF (incorporation of L-Ara4N), PagP (LPS palmitoylation), or PgaABC (deacetylation of poly-$\beta$-1,6-*N*-acetylglucosamine) (50, 51), as well as a repressor of the PhoP/PhoQ system MgrB, do not appear to have a crucial role in CSA-13 resistance. Furthermore, *E. hormaechei* adaptation to the CSA-13-induced stress was associated with exhaustive quantitative and qualitative transcriptome remodeling, including even vital ribosomal proteins. Strikingly, multiple membrane stress proteins, such as the chaperone Skp cold shock protein E (CspE), sigma factor E (RpoE), RseC, Rse, and RseP, were also downregulated by CSA-13. Subsequently, aberrantly expressed membrane-associated proteins, such as competence and type VI secretion proteins, as well as various fimbrial and flagellar components, are potential victims of this repression. Therefore, in contrast to colistin, the CSA-13-induced stress response of *E. hormaechei* is characterized by a more pronounced suppression of the acid stress response, distinctly reprogrammed metabolism, and disabling of the cell membrane protective systems.

## MATERIALS AND METHODS

**Antimicrobial agents, bacterial strains, and growth conditions.** Ceragenin CSA-13 was synthesized as previously described (52), and colistin was purchased from Sigma-Aldrich (St. Louis, MO, USA); both agents were solubilized in sterile water (molecular biology grade) (EURx, Poland). The *Enterobacter hormaechei* isolate studied was obtained from the strain collection of the Department of Medical Microbiology and Nanobiomedical Engineering (Medical University of Bialystok). The bacterial strains were grown from the freezer stocks (–80℃) on LB agar plates.

**Antimicrobial susceptibility testing.** The MICs for colistin and CSA-13 were estimated using a broth microdilution (BMD) method for serial dilutions from 128 to 0.0625 mg/L and a bacterial inoculum of $5 \times 10^5$ CFU/mL in cation-adjusted Mueller-Hinton (CAMH) medium (Sigma-Aldrich, USA). Plates were incubated overnight (18 to 20 h) at 37℃. *Escherichia coli* ATCC 25922 was used as a quality control (QC) strain.

**Induction of resistance in a serial-passage experiment.** Subinhibitory concentrations ($0.5 \times MIC$) of CSA-13 and colistin were used to induce resistance through serial passages. Briefly, during each subculture, the bacterial cells growing at the highest drug concentration were adjusted to an optical density at 600 nm ($OD_{600}$) of 0.1 and subjected to the subsequent passage at 18- to 24-h intervals. The experiment was performed for 30 consecutive days until a significant increase in MICs of $\geq$3-fold, was obtained. Furthermore, an additional 10 passages were performed to maintain the effects of induction and mimic prolonged exposure; however, there was no change in the values of the induced MICs. The experiment was conducted in triplicate for each compound.

**Nucleic acid extraction and sequencing.** DNA for WGS was isolated from overnight ($\sim$12-h) *E. hormaechei* cultures in LB with a Purify high-molecular-weight DNA kit (Promega) according to the kit protocol. Long-read sequencing was performed on the MinION apparatus (Oxford Nanopore, England) and using FlowCell 9.41 and a rapid sequencing kit. Short-read libraries (paired-end $2 \times 250$ bp) were sequenced in the Illumina NovaSeq 6000 S4 XP platform. For RNA isolation, overnight *E. hormaechei* 4236 cultures were diluted in fresh CAMH broth to an $OD_{600}$ of 0.1 and incubated at 37℃ with shaking (180 rpm) to obtain cultures at an $OD_{600}$ of 0.8 to 1.0; then, RNA was isolated from 1 mL of culture using a total RNA kit (A&A Biotechnology, Poland) according to the kit protocol. RNA samples were stored at $-80$℃. Three biological replicates were performed for the *E. hormaechei* 4236 wild type and strains resistant to colistin and CSA-13. RNA sequencing was done using a paired-end $2 \times 150$-bp read length in the Illumina NovaSeq 6000 S4 XP platform.

**Sequence processing and data analysis.** The hybrid assembly of long and short DNA reads was performed with Unicycler v.0.4. RNA reads were processed using CLC Genomics Workbench v.21.0.5. Differentially expressed genes (DEGs) with a fold change of $\geq$2.0 and $\leq$−2.0 and false-discovery rate (FDR)-adjusted *P* value of <0.05 were used for further analyses. Complete genomic sequences of the tested isolates and RNA-seq reads were deposited in the GenBank and SRA databases, respectively, as indicated below. Protein functional annotation was performed with eggNOG-mapper v.2 (53), and the DEGs were matched with metabolic pathways using Pathway Tools v.26.0 (54). Three-dimensional (3D) structures of proteins were predicted with the AlphaFold tool (55).

**Data availability.** Complete genomic sequences of the tested isolates and RNA-seq reads were deposited in the GenBank (CP104401, CP104402, CP104403, CP104404, CP104405, and CP104406 and BioProject no. PRJNA877279) and SRA (BioProject no. PRJNA881847) databases, respectively.

## SUPPLEMENTAL MATERIAL

Supplemental material is available online only.

**SUPPLEMENTAL FILE 1**, PDF file, 5.2 MB.

**SUPPLEMENTAL FILE 2**, PDF file, 1 MB.

**SUPPLEMENTAL FILE 3**, XLSX file, 0.1 MB.

## ACKNOWLEDGMENTS

This work was conducted within projects that received funding from the European Union's Horizon 2020 research and innovation program under Marie Sklodowska-Curie grant agreement no. 754432 and the Polish Ministry of Science and Higher Education, from financial resources for science in 2018 to 2023 granted for the implementation of an international cofinanced project, the Polish National Science Center (NCN)—OPUS grant (UMO-2018/31/B/NZ6/02476), and Medical University of Bialystok (SUB/1/DN/21/004/1122). The funders had no role in the design of the study, in the collection, analyses, or interpretation of data, in the writing of the manuscript, or in the decision to publish.

S.V.P. contributed Conceptualization, Writing – Original Draft; K.F. contributed Conceptualization, Writing, Visualization, Writing – Review & Editing; M.Z. contributed *Investigation*; P.B.S. contributed Writing – Review & Editing; R.B. contributed Conceptualization, Writing – Review & Editing. All authors have read and agreed to the published version of the manuscript.

The authors declare a conflict of interest. Robert Bucki received funding from N8 Medical, which focuses on commercialization of medical devices that incorporate ceragenins, to attend TechnoSepsis meeting on October 5-7, 2022 in Spain.

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
