## [Reviewer comments · Microbiology Spectrum]

Microbiology Spectrum

Glyoxylate shunt and pyruvate to acetoin shift are specific stress responses induced by colistin and ceragenin CSA-13 in *Enterobacter hormaechei* ST89

Suhanya Prasad, Krzysztof Fiedoruk, Magdalena Zakrzewska, Paul Savage, and Robert Bucki

Corresponding Author(s): Robert Bucki and Krzysztof Fiedoruk, Uniwersytet Medyczny w Białymstoku

Review Timeline:

Submission Date:	March 20, 2023
Editorial Decision:	May 12, 2023
Revision Received:	June 2, 2023
Accepted:	June 5, 2023

Editor: Emily Weinert

Reviewer(s): Disclosure of reviewer identity is with reference to reviewer comments included in decision letter(s). The following individuals involved in review of your submission have agreed to reveal their identity: Cagla Bozkurt-Güzel (Reviewer #2)

Transaction Report:

DOI: <https://doi.org/10.1128/spectrum.01215-23>

May 12, 2023

Prof. Robert Bucki
Uniwersytet Medyczny w Białymstoku
Department of Medical Microbiology and Nanobiomedical Engineering
Białystok
Poland

Re: Spectrum01215-23 (Glyoxylate shunt and pyruvate to acetoin shift are specific stress responses induced by colistin and ceragenin CSA-13 in *Enterobacter hormaechei* ST89)

Dear Prof. Bucki:

Thank you for submitting your manuscript to Microbiology Spectrum. As you will see your paper is very close to acceptance. Please modify the manuscript along the lines I have recommended. As these revisions are quite minor, I expect that you should be able to turn in the revised paper in less than 30 days, if not sooner. If your manuscript was reviewed, you will find the reviewers' comments below.

When submitting the revised version of your paper, please provide (1) point-by-point responses to the issues raised by the reviewers as file type "Response to Reviewers," not in your cover letter, and (2) a PDF file that indicates the changes from the original submission (by highlighting or underlining the changes) as file type "Marked Up Manuscript - For Review Only". Please use this link to submit your revised manuscript. Detailed instructions on submitting your revised paper are below.

Link Not Available

Sincerely,

Emily Weinert

Reviewer comments:

Reviewer #1 (Comments for the Author):

The authors selected for mutants resistant to either colistin or ceragenin CSA-13, identified mutations in resistant mutants using the whole genome sequencing, and compared gene expression profiles between the wildtype and resistant mutant strains using RNA seq. This study might provide some interesting insights into potential emergence of resistance when pathogens are exposed to antimicrobial peptides. However, there are concerns regarding the manuscript is written, in particular the result section. Authors also need to consider combining the result and discussion and changing the title. The title of the manuscript is somewhat exaggerating and speculating, because it is an authors' interpretation simply based on RNA-seq data, not experimentally proven.

In Results

The current format of the result section is difficult to follow. A brief introduction about the objective and experimental procedures in each section of the results would be helpful for readers to understand.

Ln 131-133:

It is unclear what the authors intended to mean by "in comparison."

Ln 136-137:

It is unclear what the authors intended to mean: Did it mean to say, for example, that "none of the genes in this region were previously known to be associated with colistin resistance"?

Ln 138:

- The title of this section is misleading: "Impact of colistin and CSA-13 at the gene expression level." Reading this title, I thought about gene expression in *E. hormaechei* cells grown in media containing either colistin or CSA-13. However, as I read further and only after going back to and reading the method, I realized that this experiment was to compare gene expression profiles between the wild-type strain and isolates (either resistant to colistin or CSA-13).

- In general, the authors' objective for comparing gene expression profiles between the wildtype and a resistant isolate is unclear. Any mutations (i.e., deletion, point mutation, insertion, or combination of mutations in multiple chromosomal loci) in resistant strains would be expected to affect expression of certain genes as compared with the wildtype strain. If the authors clarify the purpose of this experiment, it would be helpful for readers to understand the significance of this study.

Ln 139: "In comparison to colistin"

This is not "colistin" and, again, it is misleading. This is a colistin resistant strain.

Ln 139-140: "CSA-13 exerted more extensive dysregulation of the *E. hormaechei* transcriptome"

In general, statements like above need to be revised: for example, in the CSA-13 resistant strain, more extensive transcriptional changes were observed.

Reviewer #2 (Comments for the Author):

This manuscript focuses well on experiments and methodological part is precise

Preparing Revision Guidelines

Please return the manuscript within 60 days; if you cannot complete the modification within this time period, please contact me. If you do not wish to modify the manuscript and prefer to submit it to another journal, please notify me of your decision immediately so that the manuscript may be formally withdrawn from consideration by Microbiology Spectrum.

**Glyoxylate shunt and pyruvate to acetoin shift are specific stress responses induced by**
**colistin and ceragenin CSA-13 in *Enterobacter hormaechei* ST89**

Suhanya V. Prasad¹, Krzysztof Fiedoruk^{1*}, Magdalena Zakrzewska¹, Paul B. Savage², Robert
Bucki^{1*}

¹ Department of Medical Microbiology and Nanobiomedical Engineering, Medical University
of Bialystok, Bialystok, Poland.

² Department of Chemistry and Biochemistry, Brigham Young University, Provo, UT 84601,
USA.

*co-corresponding authors

**Corresponding Author's information:**

Krzysztof Fiedoruk (krzysztof.fiedoruk@umb.edu.pl), Robert Bucki
(buckirobert@gmail.com)

Department of Medical Microbiology and Nanobiomedical Engineering, Medical University
of Bialystok

Mickiewicza 2C Street, 15-222 Bialystok, Poland

Ph.: +48 85 784 5483

**Abstract**

Ceragenins, including CSA-13, are cationic antimicrobials that target the bacterial cell envelope
differently than colistin. However, the molecular basis of their action is not fully understood.
Here, we examined the genomic and transcriptome responses by *Enterobacter hormaechei* after
prolonged exposure to either CSA-13 or colistin. Resistance of *E. hormaechei* 4236 strain
(ST89) to colistin and CSA-13 was induced *in vitro* during serial passages with sub-lethal doses
of tested agents. The genomic and metabolic profiles of the tested isolates were characterized
using a combination of the whole genome (WGS) and transcriptome (RNA-seq) sequencing,
followed by metabolic mapping of differentially expressed genes using Pathway Tools
software. The exposure of *E. hormaechei* to colistin resulted in the deletion of the *mgrB* gene,
whereas CSA-13 disrupted the genes encoding an outer membrane protein C and transcriptional
regulator SmvR. Both compounds up-regulated several colistin-resistant genes, such as the
*arnABCDE* operon, *pagE*, including DedA proteins. The latter, along with beta-barrel protein
YfaZ and VirK/YbjX family proteins, were the top overexpressed cell envelope proteins.
Furthermore, the L-arginine biosynthesis pathway and putrescine-ornithine antiporter PotE
were down-regulated in both transcriptomes. In contrast, the expression of two pyruvate
transporters (YhjX and YjiY) and genes involved in pyruvate metabolism, as well as generating
proton motive force (PMF) genes was antimicrobial-specific. Despite the similarity of the cell
envelope transcriptomes, distinctly remodeled carbon metabolism, i.e., toward fermentation of
pyruvate to acetoin (colistin) and to the glyoxylate pathway (CSA-13), distinguished both
antimicrobials, which possibly reflects the intensity of the stress exerted by both agents.

**Keywords:**

colistin, ceragenins, CSA-13, transcriptional regulation, adaptive resistance, next-generation
sequencing, stress response, carbon metabolism

**Importance**

Colistin and ceragenins, like CSA-13, are cationic antimicrobials that disrupt the bacterial cell
envelope through different mechanisms. Here, we examined the genomic and transcriptome
changes in *Enterobacter hormaechei* ST89, an emerging hospital pathogen, after prolonged
exposure to these agents to identify potential resistance mechanisms. Interestingly, we observed
down-regulation of genes associated with acid stress response as well as distinct dysregulation
of genes involved in carbon metabolism, resulting in a switch from pyruvate fermentation to
acetoin (colistin) and the glyoxylate pathway (CSA-13). Therefore, we hypothesize that
repression of the acid stress response, which alkalinizes cytoplasmic pH and, in turn, suppresses
resistance to cationic antimicrobials, could be interpreted as an adaptation that prevents
alkalinization of cytoplasmic pH in emergencies induced by colistin and CSA-13.
Consequently, this critical for cell physiology alteration must be compensated via remodeling
carbon and/or amino acid metabolism to limit acidic byproduct production.

**Introduction**

Colistin is one of the first antibiotics discovered and is nearly as old as antimicrobial
therapy. However, its recent re-emergence as the ‘last resort’ drug against carbapenem-resistant
bacteria revealed our misunderstanding of its properties, even at the level of antibiotic
susceptibility testing (AST) (1-3).

Colistin is problematic in testing due to the variety and constantly evolving (hetero-
66)resistance mechanisms and paradoxical effects, such as ‘skipped well’ or species-dependent
increasing/decreasing MICs in the presence of divalent cations (2, 4). Therefore, recent findings
connecting resistance to colistin with globally regulated physiological processes, such as ion
homeostasis, membrane potential, respiration, and carbon metabolism, regardless of LPS

modification, e.g., by L-Ara4N, are very engaging (5-7). Notably, a study by Panta et al. (2001)
demonstrated a universal relationship between colistin activity and cytoplasmic pH
homeostasis, likely mediated by proton and non-proton pumping systems (5). Although the
molecular basis of this phenomenon remains to be elucidated, metabolic processes preventing
intracellular alkalization, such as fermentation, are certainly implicated (5, 7). Remarkably, in
this scenario, the proton motive force may directly impact interactions between colistin and
LPS due to remodeled metabolism (8).

In light of this, the purpose of the present research was to determine whether another
cationic agent – CSA-13, a member of the ceragenin family of antimicrobials, may induce
similar behaviors in bacteria. The properties of ceragenins as broad-spectrum and microbicidal
agents, along with their distinctive mode of action unrelated to cross-resistance to other
antimicrobials, are well acknowledged (9-12). Therefore, it might be assumed that the scope of
their activity comprises structures or processes universal across bacterial species. The ESKAPE
group pathogen - *Enterobacter hormaechei* ST89 was used as a target organism. The selection
is not coincidental since, in Poland, it is a growing problem, in particular in the context of
carbapenem resistance (13).

**Materials and methods**

**Antimicrobial agents, bacterial strains, and growth conditions**

Ceragenin CSA-13 was synthesized as previously described (14), and colistin was
purchased from Sigma-Aldrich (St Louis, MO, USA); both agents were solubilized in sterile
water (Molecular Biology Grade, EURx, Poland). The studied *Enterobacter hormaechei* isolate
was obtained from the strain collection of the Department of Medical Microbiology and

Nanobiomedical Engineering (Medical University of Bialystok). The bacterial strains were
grown from the freezer stocks (-80°C) on LB agar plates.

**Antimicrobial susceptibility testing**

The minimum inhibitory concentration (MIC) for colistin and CSA-13 were estimated
using a broth microdilution method (BMD) for serial dilutions from 128 to 0.0625 mg/L and
bacterial inoculum 5×10^5 CFU/mL in Cation-Adjusted Mueller Hinton medium, CAMH,
(Sigma-Aldrich, USA). Plates were incubated overnight (18-20 h) at 37 °C. *Escherichia coli*
ATCC 259222 was used as a QC strain.

**Resistance induction in serial passage experiment**

Sub-inhibitory concentrations (0.5xMIC) of CSA-13 and colistin were used to induce
resistance through serial passages. Briefly, during each subculture, the bacterial cells growing
at the highest drug concentration were adjusted to OD₆₀₀ 0.1 and subjected to the following
passage at 18-24 h intervals. The experiment was performed for 30 consecutive days until a
significant increase in MICs ≥ 3 -fold, was obtained. Furthermore, additional 10 passages were
performed, to maintain the effects of induction and mimic prolonged exposure; however, there
was no change in values of the induced MICs. The experiment was conducted in triplicates for
each compound.

**Nucleic acid extraction and sequencing**

DNA for WGS was isolated from overnight (~12 h) *E. hormaechei* cultures in LB with
Purify High-Molecular-Weight DNA kit (Promega) according to the kit protocol. Long-read
sequencing was performed on the MinION apparatus (Oxford Nanopore, England) and using
FlowCell 9.41 and Rapid Sequencing Kit. Short-read libraries (paired-end 2x250 bp) were

sequenced in the Illumina NovaSeq 6000 S4 XP platform. For RNA isolation overnight *E.*
*hormaechei* 4236 cultures were diluted in fresh CAMH broth to OD₆₀₀ 0.1 and incubated at
37°C with shaking (180 RPM) to obtain cultures at density OD₆₀₀ 0.8-1.0; then RNA was
isolated from 1 mL of culture using Total RNA kit (A&A Biotechnology, Poland) according to
the kit protocol. RNA samples were stored at -80°C. Three biological replicates for the *E.*
*hormaechei* 4236 wild type and resistant to colistin and CSA-13 were performed. RNA
sequencing was done using a paired-end 2x150 bp read length in the Illumina NovaSeq 6000
S4 XP platform.

Sequences processing and data analysis

The hybrid assembly of long and short DNA reads was performed with Unicycler v0.4.
RNA reads were processed using CLC Genomics Workbench v 21.0.5. Differentially expressed
genes (DEGs) with a fold-change ≥ 2.0 and ≤ -2.0 and FDR-adjusted p-value of <0.05 were
used for further analyses. Complete genomic sequences of the tested isolates and RNA-seq
reads were deposited in the GenBank (CP104401, CP104402, CP104403, CP104404,
CP104405, and CP104406; BioProject PRJNA877279) and SRA (BioProject PRJNA881847)
databases, respectively. Protein functional annotation was performed with eggNOG-mapper v2
(15), and the DEGs were matched with metabolic pathways using Pathway Tools v26.0 (16).
3D structures of proteins were predicted with AlphaFold tool (17).

Results

Induced resistance to colistin and CSA-13

Resistance induction in *E. hormaechei* 4236 strain to colistin and CSA-13 by serial
passage experiments resulted in a high level of resistance to colistin, i.e., an increase in MIC
from 0.125 mg/L to 128 mg/L (denoted hereinafter as Eh4236ColR isolate). In contrast, the

final MIC for CSA-13 varied between 32 to 64 mg/L (denoted hereinafter as Eh4236Csa13R

[revised manuscript text omitted]

**Author Contributions**

Conceptualization, writing, and original draft preparation, S.V.P.; conceptualization, writing,
figure preparation, and review and editing, K.F.; participation in *in vitro* experiments, M.Z.;
review and editing, P.B.S.; conceptualization and review and editing, R.B. All authors have
read and agreed to the published version of the manuscript.

**Funding**

This work was conducted within projects which received funding from the European Union's
Horizon 2020 research and innovation program under the Marie Skłodowska-Curie grant
agreement No. 754432 and the Polish Ministry of Science and Higher Education, from financial
resources for science in 2018–2023 granted for the implementation of an international co-
financed project, the Polish National Science Center (NCN) – OPUS grant (UMO-
2018/31/B/NZ6/02476) and Medical University of Białystok (SUB/1/DN/21/004/1122).

**Conflicts of Interest**

The authors declare no conflict of interest. The funders had no role in the design of the study,
in the collection, analyses, or interpretation of data; in the writing of the manuscript; or in the
decision to publish.

**Data Availability**

Complete genomic sequences of the tested isolates and RNA-seq reads were deposited in the
GenBank (CP104401, CP104402, CP104403, CP104404, CP104405, and CP104406;
BioProject PRJNA877279) and SRA (BioProject PRJNA881847) databases, respectively.

**Supplemental material**

Supplemental material is available online only.

SUPPLEMENTAL FILE 1, Tables S1 – S4, XLSX file, 0.09 MB.

SUPPLEMENTAL FILE 2, Figures S1 – S2, PDF file, 39 MB.

SUPPLEMENTAL FILE 3, Figures S3 – S15, PDF file, 0.9 MB

[revised manuscript text omitted]

- 567

June 5, 2023

Prof. Robert Bucki
Uniwersytet Medyczny w Białymstoku
Department of Medical Microbiology and Nanobiomedical Engineering
Białystok
Poland

Re: Spectrum01215-23R1 (Glyoxylate shunt and pyruvate to acetoin shift are specific stress responses induced by colistin and ceragenin CSA-13 in *Enterobacter hormaechei* ST89)

Dear Prof. Bucki:

Your manuscript has been accepted, and I am forwarding it to the ASM Journals Department for publication. You will be notified when your proofs are ready to be viewed.

Sincerely,

Emily Weinert
Editor, Microbiology Spectrum